# Inhibition of Prolactin Affects Epididymal Morphology by Decreasing the Secretion of Estradiol in Cashmere Bucks

**DOI:** 10.3390/ani14121778

**Published:** 2024-06-13

**Authors:** Xiaona Liu, Chunhui Duan, Xuejiao Yin, Lechao Zhang, Meijing Chen, Wen Zhao, Xianglong Li, Yueqin Liu, Yingjie Zhang

**Affiliations:** 1College of Animal Science and Technology, Hebei Agricultural University, Baoding 071001, China; liuxiaonahau@163.com (X.L.); duanchh211@126.com (C.D.); 18531132767@163.com (L.Z.); chenmeijing815@126.com (M.C.); zhaowen970920@163.com (W.Z.); 2College of Animal Science and Technology, Hebei Normal University of Science & Technology, Qinhuangdao 066004, China; bdyinxuejiao@foxmail.com (X.Y.); 15203358192@163.com (X.L.)

**Keywords:** prolactin, epididymis, reproduction, buck, PRLR

## Abstract

**Simple Summary:**

Abnormal prolactin levels can lead to male infertility. However, the regulation mechanism of prolactin in the epididymis is still unclear. We studied the effects of prolactin on the epididymis function of Yanshan Cashmere bucks using the prolactin inhibitor bromocriptine. The results show that the inhibition of prolactin decreased the serum estradiol concentrations and the expression of the prolactin receptor protein in the epididymis (*p* < 0.05). The inhibition of prolactin also increased the height of the epididymal epithelium in the caput and cauda, as well as the diameter of the epididymal duct in the caput (*p* < 0.05). Meanwhile, prolactin inhibition decreased the duct diameter in the epididymal cauda (*p* < 0.05). Then, transcriptome analyses showed that the differentially expressed genes *ESR2*, *MAPK10*, *JUN*, *ACTL7A*, and *CALML4* were enriched in multiple pathways, including the estrogen signaling pathway, the GnRH signaling pathway, the cAMP signaling pathway, the chemical carcinogenesis–reactive oxygen species pathway, steroid binding, and calcium ion binding, which may be the key genes in the prolactin regulation of epididymal function. This study provides new insights into the molecular mechanisms of prolactin regulating epididymis function in bucks.

**Abstract:**

Yanshan Cashmere bucks are seasonal breeding animals and an important national genetic resource. This study aimed to investigate the involvement of prolactin (PRL) in the epididymal function of bucks. Twenty eleven-month-old Cashmere bucks were randomly divided into a control (CON) group and a bromocriptine (BCR, a prolactin inhibitor, 0.06 mg/kg body weight (BW)) treatment group. The experiment was conducted from September to October 2020 in Qinhuangdao City, China, and lasted for 30 days. Blood was collected on the last day before the BCR treatment (day 0) and on the 15th and 30th days after the BCR treatment (days 15 and 30). On the 30th day, all bucks were transported to the local slaughterhouse, where epididymal samples were collected immediately after slaughter. The left epididymis was preserved in 4% paraformaldehyde for histological observation, and the right epididymis was immediately preserved in liquid nitrogen for RNA sequencing (RNA-seq). The results show that the PRL inhibitor reduced the serum PRL and estradiol (E2) concentrations (*p* < 0.05) and tended to decrease luteinizing hormone (LH) concentrations (*p* = 0.052) by the 30th day, but no differences (*p* > 0.05) occurred by either day 0 or 15. There were no differences (*p* > 0.05) observed in the follicle-stimulating hormone (FSH), testosterone (T), and dihydrotestosterone (DHT) concentrations between the two groups. The PRL receptor (PRLR) protein was mainly located in the cytoplasm and intercellular substance of the epididymal epithelial cells. The PRL inhibitor decreased (*p* < 0.05) the expression of the PRLR protein in the epididymis. In the BCR group, the height of the epididymal epithelium in the caput and cauda increased, as did the diameter of the epididymal duct in the caput (*p* < 0.05). However, the diameter of the cauda epididymal duct decreased (*p* < 0.05). Thereafter, a total of 358 differentially expressed genes (DEGs) were identified in the epididymal tissues, among which 191 were upregulated and 167 were downregulated. Gene Ontology and Kyoto Encyclopedia of Genes and Genomes analyses revealed that *ESR2*, *MAPK10*, *JUN*, *ACTL7A*, and *CALML4* were mainly enriched in the estrogen signaling pathway, steroid binding, calcium ion binding, the GnRH signaling pathway, the cAMP signaling pathway, and the chemical carcinogenesis–reactive oxygen species pathway, which are related to epididymal function. In conclusion, the inhibition of PRL may affect the structure of the epididymis by reducing the expression of the PRLR protein and the secretion of E2. *ESR2*, *MAPK10*, *JUN*, *ACTL7A*, and *CALML4* could be the key genes of PRL in its regulation of epididymal reproductive function.

## 1. Introduction

The epididymis plays a vital role in ensuring proper sperm maturation and enabling male reproductive function [1,2]. Epididymal function depends on the highly specialized epididymal luminal microenvironment, in which sperms mature and are protected [1,2,3]. Due to the complex functions and compartmentalization of the epididymis, multiple causes of epididymal dysfunction leading to male fertility disorders and even healthy offspring are conceivable [4,5]. Although male infertility is common, it affects about 15% of couples of child-bearing age worldwide, and more than 50% of cases are idiopathic [6,7,8]. The complex network of regulatory mechanisms involved in epididymal function still needs to be further studied.

Prolactin (PRL) is a polypeptide hormone synthesized and secreted by the anterior pituitary. Its receptors are widely found in many tissues and organs [9]. After binding to the PRL receptor (PRLR) on the surface of cell membrane, PRL exerts its biological function by initiating the corresponding intracellular signal transduction process [10]. Many studies have shown that PRL is involved in many physiological processes of vertebrate reproduction, osmoregulation, growth and development, metabolism, immune regulation, energy balance, and behavior regulation [11]. Although male animals cannot lactate, their plasma levels of PRL are similar to those of females, suggesting that PRL may influence male reproductive function [11]. Meanwhile, studies have found that PRL receptors are expressed in the epididymis of various animals, including male rats, male mice, rabbits, red deer, and bovines [12,13,14,15,16], suggesting that PRL plays an important role in the epididymis of these animals.

Several studies have shown that an abnormal PRL concentration can affect reproductive function. Clinical observations in infertile men with hypoprolactinemia have demonstrated that the restoration of normal PRL concentrations results in an increase in sperm density and quality and a restoration of fertility, suggesting a regulatory effect of PRL on the testis and epididymis [17]. However, the PRL concentrations in the blood circulation of males are higher than the reference range of the normal population, which may decrease spermatogenesis and lead to male infertility [18]. Studies on PRL in mice have shown inconsistent results. When the PRLR was knocked out or PRLR function was blocked, half of the male mice were found to be infertile or had reduced fertility [19]. But when targeted disruption of the PRLR gene in mice was observed, no impairment of their testicular function and fertility was found [20]. In red deer, PRLR gene expression was extremely high during the breeding period and decreased significantly during the non-breeding period [21]. During the breeding season, PRLR expression levels were higher in the epididymis of red deer than in the interstitial and seminiferous tubule compartments of the testis [14]. Thus, studies on the effects of PRL on male fertility have shown inconsistent results among different animals.

Yanshan Cashmere goats are seasonal breeding animals and were identified as a national genetic resource in 2021 in China. They have been bred for more than 600 years, and they are characterized by their high cashmere production, high-quality mutton, disease resistance, and adaptability. They are mainly found in the Yanshan area of Hebei Province, China, and their breeding scale is about 2 million. PRL showed seasonal variation in Cashmere goats [22]. Based on previous findings, we hypothesized that PRL may play an important role in the regulation of epididymal physiology in bucks. Therefore, this experiment was designed to study the effects of PRL inhibition on the reproductive hormones, epididymal morphology, and reproduction-related genes in Cashmere bucks, the results of which may provide a theoretical basis for revealing the regulatory mechanisms of PRL in regulating epididymal function in male animals.

## 2. Materials and Methods

### 2.1. Ethics Statement

Our study was conducted under the guidance of the Animal Care and Use Committee of Hebei Agricultural University (approval number: 2024140).

### 2.2. Animals and Management

This study was conducted from September to October 2020 at the Qinglong Lihong Cashmere Goat Farm, Qinhuangdao City, China (longitude 118°95′ E, latitude 40°41′ N, and at an altitude of 245 m). During the experimental period, all bucks were in a natural photoperiod (~11.7 h of daylight) and had free access to fresh water. The bucks were fed twice a day (07:00 and 17:00), with a mixture of corn stalks, corn, and concentrate supplements. The feeding management of the experimental bucks was in step with the goat farm. Ivermectin was used for deworming before the start of the experiment. The diets provided for bucks included 48.7% corn stalks, 37.8% corn, and 13.5% concentrate (crude protein 30%, crude ash 30%, crude fiber 12%, Ca 1.5% to 4.5%, NaCl 1.0% to 4.0%, Lys 1.2%, total phosphorus 1.0%, and moisture 13.0%, purchased from the Chaoyang Tianqin Feed Company, Chaoyang, China). The dietary metabolizable energy rate was 8.5 MJ/kg, and the crude protein level was 9.5%.

### 2.3. Experimental Design and Sample Collection

Twenty eleven-month-old male Yanshan Cashmere bucks with a live weight of 23.91 ± 0.82 kg were randomly assigned to two groups (n = 10): control (CON) and bromocriptine treatment (BCR). The bucks in the BCR group were treated with bromocriptine (Gedeon Richter Plc, Budapest, Hungary), which was dissolved in water and sprayed evenly into the feed so that each animal received approximately 1.5 mg per day for the duration of the experiment. The bucks in the CON group were sprayed with the same amount of water. The dose of bromocriptine (0.06 mg/kg body weight, BW) used in this study was based on the studies by Dicks et al. [23], Zhang et al. [24], and the BCR instructions (about 0.05 to 0.07 mg/kg). The experiment lasted for 30 days.

Blood was collected from the bucks on the 10th (day 0) and 25th (day 15) of September and the 10th (day 30) of October 2020 at 7:30 a.m. using a 5 mL vacuum procoagulant tube. The blood samples were immediately centrifuged at 1200× *g* for 10 min, and the resulting serum samples were stored at −20 °C. After the blood collection on 10 October 2020, all bucks were transported to the local slaughterhouse, where epididymal samples were collected immediately after slaughter. The left epididymal tissue was immersed in 4% paraformaldehyde for subsequent experiments with histological analyses. The right epididymal caput tissue was collected and immediately frozen in liquid nitrogen for subsequent RNA sequencing.

### 2.4. Hormone Analyses

The serum levels of PRL (H095), estradiol (E2, H102), luteinizing hormone (LH, H206), follicle-stimulating hormone (FSH, H101), testosterone (T, H090), and dihydrotestosterone (DHT, H293) were analyzed using commercial goat ELISA kits provided by the Nanjing Jiancheng Bioengineering Institute (http://www.njjcbio.com/ accessed on 10 October 2020). The goat ELISA kit for PRL, E2, LH, FSH, and T was batch number 20201010. The goat ELISA kit for DHT was batch number 20230320. Serum samples were obtained from 20 Cashmere bucks. This test kit applies the competition method to detect the contents of hormones. The sensitivity of each assay was 0.5 ng/mL (PRL), 2 ng/L (E2), 0.1 mIU/mL (LH), 0.2 mIU/mL (FSH), 0.05 ng/L (T), and 10 ng/L (DHT). The intra-assay coefficients of variation were <10%. Hormone data were obtained from standard curves by fitting a logistic (four-parameter) curve model in ELISAcalc software (v 0.1). The fit indicators refer to the R^2^ of the standard curves, which are all above 0.99. Each hormone analysis was performed in duplicate.

### 2.5. Hematoxylin–Eosin Staining and Immunohistochemistry of Epididymal Tissue

The epididymal tissue collected from 16 bucks was fixed in 4% paraformaldehyde and stored at room temperature overnight. The fixed tissue was dehydrated using a fully automated dehydrator in a gradient ethanol bath of increasing concentrations (75, 85, 95, and 100%). The tissue was then infiltrated using paraffin. Each paraffin block was serially sectioned using a rotary slicer (Leica-2016, Wetzlar, Germany), and the slices were 5 μm thick. For hematoxylin–eosin (HE) staining, some sections were deparaffinized with xylene and ethanol. Afterward, they were stained with HE, dehydrated, and sealed with neutral gum.

A Panoramic 250 digital section scanner from 3DHISTECH (Budapest, Hungary) was used for image acquisition of the sections. Each section of tissue was first observed at 40× magnification to observe the general lesions, and 100×, 200×, and 400× magnification images were acquired to observe areas of specific pathological changes. Histopathological analyses were reported by two pathologists who were unaware of the group identities. Measurements were performed with the measurement tool in CaseViewer image analysis software (2.3.0.99276), and 30 epididymal ducts most resembling a circle were collected from each sample to measure the epididymal duct diameter, epididymal lumen diameter, and epithelium height [25,26].

In addition, five paraffin-embedded sections of the epididymal caput of bucks were randomly selected from each group for PRLR immunohistochemistry. Five-micrometer-thick sections were deparaffinized with xylene and ethanol. To expose the epitopes, the sections were subjected to microwave treatment in citrate buffer at pH 6.0, cooled, and washed with PBS. The sections were then incubated in 0.3% hydrogen peroxide for 10 min at room temperature to destroy endogenous peroxidase activity. After washing with PBS, the sections were blocked with 1% bovine serum albumin, after which primary antibodies (NB300-561, 1:100, Novus Biologicals, Centennial, CO, USA) were added, and they were incubated overnight at 4 °C. The sections were washed with PBS, then secondary antibodies (GB23301, 1:100, Servicebio, Wuhan, China) were added, and they were incubated for 30 min at 37 °C. The sections were washed again with PBS, freshly prepared diaminobenzidine chromogenic solution was added to the tissue, and the color was developed at room temperature. Under the microscope, the color development time was controlled; positive staining was yellowish brown. The color development was terminated after washing the sections with distilled water. Subsequently, we counterstained the nuclei with hematoxylin for 3 min and finally dehydrated and sealed the slides with neutral gum. The primary antibody in the negative control experiments was replaced with PBS [13,26]. We used a microscopic camera system (BA400Digital, Mc Audi Industries Group Ltd., Beijing, China) to acquire images of the sections, and each section of all tissue was first observed at 100× magnification, and then microscopic images were obtained at 100× and 400× magnifications. The percentage of positive area (%DAB-positive tissue) was calculated for each 400× image using the Halo data analysis system (Halo 101-WL-HALO-1, Indica Labs, Albuquerque, NM, USA).

### 2.6. Library Preparation, RNA Sequencing, and Bioinformatics Analyses

Three samples were randomly selected from each group, and the total RNA was isolated from the epididymal caput tissues of the CON and BCR groups using TRIzol Reagent (Invitrogen Life Technologies, Carlsbad, CA, USA), after which the concentration, quality, and integrity were determined using a NanoDrop spectrophotometer (Thermo Scientific, Waltham, MA, USA). RNA sequencing (RNA-seq) libraries were prepared from the total RNA (2 μg), and the ribosomal RNA was removed using the Epicentre Ribo-Zero™ rRNA Removal Kit. After obtaining the libraries, library quality testing was performed using an Agilent 2100 Bioanalyzer (Agilent, 2100; Santa Clara, CA, USA) and Agilent High Sensitivity DNA Kit (Agilent, 5067-4626). The sequencing library was then sequenced on the NovaSeq 6000 platform (Illumina, San Diego, CA, USA) by Shanghai Paisano Biotechnology Co., Ltd. (Shanghai, China).

The quality information was calculated for the raw data in FASTQ format, including the sample name, Q20 (%), Q30 (%), etc. Clean data were obtained after filtering the raw data using Cutadapt (v1.15) software. The reference genome and gene annotation files of Capra hircus were downloaded from the genome website. All subsequent analyses were based on high-quality clean data. Filtered reads were mapped to the reference genome using HISAT2 v2.0.5. We used HTSeq (v0.9.1) to statistically compare the read count values on each gene as the original expression of the gene, and then used FPKM to standardize the expression. DESeq (v1.38.3) was used to analyze differentially expressed genes. The screening conditions were the following: expression difference multiple |log2FoldChange| > 1, significant *p*-value < 0.05. Meanwhile, the P heatmap (v1.0.12) package of R language was used to perform bidirectional cluster analysis of different genes in all samples. Heatmaps were drawn based on the expression levels of the same gene in different samples and the expression patterns of different genes in the same sample, with the Euclidean method used to calculate the distance and the complete linkage method used for clustering.

We mapped all the genes to terms in the Gene Ontology (GO) database and calculated the numbers of differentially enriched genes in each term. GO enrichment analysis of differential genes (all DEGs/upper DEGs/lower DEGs) was performed using topGO (v2.50.0). The *p*-value was calculated using the hypergeometric distribution method (the standard of significant enrichment was a *p*-value < 0.05). The GO terms of significantly enriched differential genes were found, and the main biological functions of differential genes were determined. Cluster Profiler (v4.6.0) software was used to carry out the enrichment analysis of the Kyoto Encyclopedia of Genes and Genomes (KEGG) pathway of differential genes, focusing on the significant enrichment pathway with a *p*-value < 0.05.

### 2.7. Quantitative Real-Time PCR

The 20 selected transcriptome DEGs were validated using quantitative real-time PCR (qPCR). The total RNA from the epididymal tissue was extracted using TRIzol™ Reagent (Invitrogen, USA), and the total RNA (1 μg) tested was reverse-transcribed into cDNA according to the kit instructions (Prime Script TM 1st stand cDNA Synthesis Kit). The reverse transcription reaction system included 1 μL of oligo (dT), 1 μL of dNTP mix (10 mmol/L), 4 μL of 5× reaction buffer, 0.5 μL of RNase inhibitor (40 U/μL), 1 μL of MMLVRT (200U/μL), and RNase-free dH_2_O up to 20 μL. The primers used for qPCR are listed in Table 1 and were synthesized by Shanghai Saiheng Biotechnology Co., Ltd. (Shanghai, China). The PCRs were performed using the AceQ^®^ qPCR SYBR^®^ Green Master Mix (Vazyme, China) on a real-time PCR instrument (MA-6000, Suzhou Yarrow Biotechnology Co., Ltd., Suzhou, China). The qPCR system included 10 μL of 2× SYBR real-time PCR premixture, 0.4 μL of 10 μM PCR-specific primer F, 0.4 μL of 10 μM PCR-specific primer R, 1 μL of cDNA, and RNase-free dH_2_O up to 20 μL. The amplification cycling conditions were as follows: 95 °C for 5 min (pre-denaturation), followed by 40 cycles of 95 °C for 15 sec (denaturation), and 60 °C for 40 s (annealing and extension). The reaction ended and the data were obtained. The specificity of each primer set was tested using melting curve analysis for all genes in the range of 60 to 95 °C. The experiment was repeated three times. Data analyses were performed using the 2^−∆∆Ct^ method. *β-actin* was used as a reference gene for the relative quantitative real-time PCR experiments.

### 2.8. Statistical Analyses

Data from the CON and BCR groups were statistically analyzed using the Student’s t-test procedure with SPSS 21.0 statistical analysis software. A *p* < 0.05 was considered statistically significant. The data were expressed as the mean ± standard error of the mean (SEM).

## 3. Results

### 3.1. Effects of BCR Treatment on Serum Hormone Levels in Cashmere Bucks

Compared to the CON group, the inhibition of PRL decreased the levels of serum PRL and E2 by day 30 (*p* < 0.05; Figure 1A,B). The levels of LH in the BCR group tended to decrease by day 30 (*p* = 0.052; Figure 1D). No changes in these hormones occurred by days 0 or 15 (*p* > 0.05; Figure 1). Prolactin inhibition had no effect on the serum FSH, T, or DHT levels during the whole experimental period (*p* > 0.05; Figure 1C,E,F).

### 3.2. Effects of BCR Treatment on Protein Expression of PRLR in Epididymal Tissue of Cashmere Bucks

It was found that PRLR was localized in the cytoplasm and intercellular substance of the epididymal epithelial cells of Yanshan Cashmere bucks (Figure 2A–D). The nuclei stained with hematoxylin were blue, and the DAB-positive expression was yellowish brown. The negative control section is shown Figure 2E. The %DAB-positive tissue of PRLR was decreased in the BCR group compared to the CON group (Figure 2F, *p* < 0.05).

### 3.3. Effects of BCR Treatment on the Histomorphology of the Epididymis in Cashmere Bucks

The results of the HE-stained sections of the epididymis show that there were no significant pathological changes in the histomorphology of the caput and cauda of the epididymis in either the CON or BCR group (Figure 3). The diameter of the epididymal duct was smaller in the caput than that in the cauda, and the epithelium height was larger in the caput than that in the cauda. Compared to the CON group, statistical analysis of the measured data shows that the BCR group in the caput of the epididymis increased the epididymal duct diameter (302.83 ± 2.88 μm vs. 282.74 ± 2.53 μm, *p* < 0.05), epididymal lumen diameter (186.00 ± 3.15 μm vs. 171.48 ± 2.60 μm, *p* < 0.05), and epididymal epithelium height (61.19 ± 0.69 μm vs. 58.82 ± 0.57 μm, *p* < 0.05; Figure 3A–D,I). In the cauda of the epididymis, the BCR group decreased the diameter of the epididymal duct (589.36 ± 12.14 μm vs. 623.97 ± 8.23 μm, *p* < 0.05) and the lumen of the epididymis (542.62 ± 13.19 μm vs. 582.81 ± 8.58 μm, *p* < 0.05)), but increased the height of the epididymal epithelium in the cauda of the epididymis (23.17 ± 0.65 μm vs. 20.70 ± 0.40 μm, *p* < 0.05; Figure 3E–H,J).

### 3.4. RNA Sequencing Data Statistics and Differentially Expressed Gene Analyses

We analyzed the RNA sequencing data from the CON and BCR groups and obtained 146,488,370–162,455,762 raw reads and 133,759,152–144,256,712 clean reads (Table 2). A total of 358 differentially expressed genes (DEGs) were identified using DESeq, including 191 upregulated genes and 167 downregulated genes (Figure 4A). Figure 4B shows the hierarchical clustering of the differentially expressed (DE) mRNAs.

### 3.5. Enrichment and Functional Annotation Analyses of DEGs

GO enrichment analyses of DEGs were classified according to biological process (BP), cellular component (CC), and molecular function (MF). The top 10 GO term entries with the lowest *p*-value, the most significant enrichment, were selected from each GO classification for display (Figure 5A). We found that these DEGs (including *ESR2*, *GPER1*, *ACTL7A*, *STPG4*, *CYP2R1*, *GC*, and *CALML4*) were significantly enriched in the male germ cell nucleus, estrogen receptor activity, steroid binding, D3 vitamins binding, and calcium ion binding. To further evaluate the role of PRL inhibition in epididymal reproductive physiological pathways, we performed KEGG pathway enrichment analysis using clusterProfiler (v4.6.0) software and found that 239 KEGG pathways were enriched. We selected the top 20 KEGG pathways with the smallest FDR values, which indicated the most significant enrichment, as shown in the bubble plot (Figure 5B). The results show that these DEGs (including *ESR2*, *MAPK10*, *JUN*, *CALML4*, and *ADCY10*) were significantly enriched in the estrogen signaling pathway, GnRH signaling pathway, cAMP signaling pathway, chemical carcinogenesis–reactive oxygen species, and phosphatidylinositol signaling system pathway. These pathways were associated with epididymal homeostasis and sperm maturation.

### 3.6. Validation of Differentially Expressed mRNAs by qPCR

To validate our RNA-seq results, 20 differentially expressed mRNAs were identified, including *ACRV1*, *SPESP1*, *ACTL7A*, *MAPK10*, *PLA2G4E*, *ACTRT3*, *ADCY10*, *ESR2*, *STPG4*, *FAM170B*, *CYP2R1*, *GC*, *ENPP3*, *JUN*, *CALML4*, *GPER1*, *POMC*, *ENSCHIG00000016126*, *ENSCHIG00000024353*, and *ENSCHIG00000026189*. As shown in Figure 6, the relative fold changes in the qPCR assay are consistent with the RNA-seq results, indicating the reliability of our RNA-seq data.

## 4. Discussion

In the present study, we hypothesized that the inhibition of PRL secretion would have an effect on epididymal reproductive function in bucks. The results showed that the inhibition of PRL secretion changed the serum hormone level, PRLR protein expression, and epididymis morphology and gene expression level of epididymis, which were consistent with our hypothesis.

### 4.1. Effects of BCR Treatment on Serum Hormone Concentrations

Numerous studies have shown that abnormal PRL concentrations affect reproductive function in animals [17,18]. In this study, the inhibition of PRL secretion using BCR resulted in some changes in the serum hormone levels, histomorphology of the epididymis, prolactin receptor protein expression, and transcriptome results in Cashmere bucks, which are consistent with our expected results. BCR is a dopamine D2 receptor agonist that has been widely used to study the physiologic effects of PRL [27]. Studies on red deer and Scottish blackface ewes have shown that BCR failed to sustain the long-term suppression of PRL secretion [28,29]. Long-acting BCR treatment of red deer resulted in serum PRL concentrations that were first unchanged, then decreased, and finally returned to the control levels [28]. The duration of this period showed differences in length depending on the species. The reason for the failure to maintain inhibition over the long term may be that dopamine acts primarily as a modulator of short-term PRL release rather than syn-thesis [30]. This is in contrast to photoperiod-induced PRL suppression, which inhibits PRL synthesis and release [30]. The same phenomenon was observed in the present experiment, where PRL levels were unchanged after 15 days of BCR treatment, whereas PRL levels were significantly decreased after 30 days of BCR treatment. Meanwhile, we found that when the prolactin levels were decreased, other hormones also changed. Prolactin is related to estrogen, and studies have shown that estrogen stimulates pituitary prolactin production and secretion through the inhibition of hypothalamic dopamine [31,32]. This may explain the decrease in serum E2 after BCR treatment in our study. In general, estrogen stimulates prolactin secretion, and high prolactin concentrations inhibit estrogen secretion [32]. However, in our experiments, when prolactin levels were low, estradiol was also decreased, which may be related to the fact that our experimental animals were bucks. Studies have found that estrogens also regulate male reproduction [33]. Aromatase converts androgens to estrogens and has been reported to be present in the epididymal epithelium and mesenchyme to provide estrogen when sperm are not the primary source of estrogen [34,35,36,37]. Thus, when E2 decreases with decreasing levels of PRL, there is an effect on the reproductive function of the buck’s epididymis. In similar experiments, BCR treatment in adult rats resulted in a significant decrease in LH binding in the testes [38], and BCR also decreased LH receptors in the testes of rams [39]. This suggests that PRL plays an important role in the maintenance of LH receptors. This could explain why the serum LH concentration was lower in the BCR group in our study. The above study suggests that PRL plays an important role in maintaining LH receptors. This could be the reason for the lower LH concentration in the BCR group in the experiment.

### 4.2. Effects of BCR Treatment on the PRLR Protein Expression and Histomorphology in the Epididymis

Consistent with the results of most animal studies (including rat, mouse, rabbit, red deer, and bovine) [12,13,14,15,16], our results confirm that PRLR was also present in the epididymis of Yanshan Cashmere bucks, and mainly present in the cytoplasm and intercellular substance of the epididymis epithelium. BCR treatment also reduced PRLR protein expression in the epididymal tissue. This illustrates that PRLR plays an important role in epididymal epithelial cells. Epididymal function depends on the highly specialized epididymal microenvironment, which is formed and maintained by the activity of epididymal epithelial cells [2]. During the breeding season, PRLR expression levels were higher in the epididymis of red deer than in the interstitial of the testes [14]. This demonstrates the importance of PRL in the epididymal reproductive function of seasonal breeding animals.

PRL is also believed to maintain epididymal weight in breeding season mice, stimulate the absorption of testosterone, and promote fluid exchange between the epididymal epithelium [13]. The chronic inhibition of PRL also led to reductions in the seminal vesicle size and fructose content in rams [40]. We found that the inhibition of PRL did not cause significant pathological changes in the epididymal tissue, but the diameter of the epididymal duct and the height of the epididymal epithelium were significantly changed. The structure and function of the caput and cauda of the epididymis are specific [41,42]. The epididymal epithelium height in the caput is higher than that in the cauda, and it mainly performs the role of reabsorption in the epididymis, absorbing fluid from the testicular network [43]. The epididymal caudal lumen diameter is larger than that of the caput, and the cauda performs the functions of sperm maturation and the concentration of stored fluids [2,43]. Thus, the different effects of PRL inhibition on the lumen diameters of the caput and cauda are related to the fact that they do not exercise the same functions. The results of the above study show that the BCR treatment affected the expression of the PRLR protein in the epididymis and altered the morphological structure of epididymal tissue.

### 4.3. Effects of BCR Treatment on Epididymal Reproductive Function-Related Genes

The maintenance of epididymal function is a complex network of molecular regulation. Understanding their functional differences at the molecular level could better elucidate the changes in hormone levels and epididymal tissue morphology caused by prolactin inhibition in bucks. Studies have shown that estrogens and their major nuclear receptors (ESR1 and ESR2) and the G protein-coupled estrogen receptor (GPER) also regulate male reproduction [33]. In fat sand rats, *GPER1* mediates specific cellular estrogen signaling, and testosterone is involved in regulating *GPER1* expression, in addition to other estrogen signaling pathways [44]. When JNK and c-Jun are associated with the cAMP signaling pathway, they lead to increased steroidogenic gene expression [45]. We identified the significant enrichment of estrogen receptor activity, steroid binding, the estrogen signaling pathway, the cAMP signaling pathway, the GnRH signaling pathway, etc., using GO and KEGG analyses. Major differential genes, such as *ESR2*, *GPERI*, *JUN*, *ADCY10*, *PLA2G4E*, and *MAP2K6*, were significantly enriched in these pathways. These genes’ changes might explain the lower E2 concentration due to PRL inhibition.

Epididymal epithelial structural alterations or the obstruction of the epididymal lumen may lead to male infertility, so the normal maintenance of epididymal epithelial cell morphology and function is essential for the immune defense of the blood–epididymis barrier [46]. In the present study, we found that DEGs (including *ACTL7A, ACTRT3,* and *SPESP1*) were significantly enriched in cell components, such as the germ cell nucleus and male germ cell nucleus. *ACTL7A* is essential for early embryonic development, and its reduction could lead to poor embryonic development and even infertility [47,48,49,50]. The deletion of *SPESP1* led to an abnormal distribution of multiple proteins in mouse spermatozoa, reduced the sperm fusion capacity, and reduced the number of pups [51]. *ACTRT3* belongs to the Actin-related proteins family. It is localized in the nucleus of male germ cells and is involved in chromatin and nuclear structure, as well as cytoplasmic functions [52]. These differential genes related to cell components might have an important relationship with the morphological changes in the epididymis. The differentially upregulated gene *MAPK10*, a member of the MAP kinase family, is involved in a variety of cellular biological processes, such as proliferation, differentiation, and development [53]. The increased height of epididymal epithelial cells might be associated with alterations in this gene.

Changes in the calcium concentrations, AMP concentrations, sperm intracellular pH, and phosphatase and kinase activities during epididymal transit are important factors related to sperm maturation and the acquisition of motility [54,55]. The COOH terminal of PRLR’s intracellular domain is involved in the production of the intracellular messengers that open voltage-independent calcium ion channels [56,57]. This study found that the inhibition of PRL upregulated the *CYP2R1* and *GC* genes and downregulated the *CALML4* gene. *CYP2R1* is a cytochrome P450 monooxygenase involved in the activation of vitamin D precursors [58,59]. *GC* is involved in vitamin D transport and storage [60,61,62]. *CALML4* belongs to the calmodulin superfamily, which is known to interact with a diverse set of target proteins that function in numerous cellular pathways [63]. *CALML4* is predicted to enable calcium ion binding activity and enzyme regulatory activity. Based on the changes in these genes, we suggest that PRL might affect the calcium homeostasis in the microenvironment of the epididymis. Interestingly, we also found that differentially expressed genes inhibiting PRL production were significantly enriched in the cAMP signaling pathway, phosphatidylinositol signaling system–reference pathway, and chemical carcinogenesis–reactive oxygen species pathway (ROS). The changes in the cAMP contents and phosphodiesterase concentration in the epididymis will affect the motility of sperm [54]. An appropriate amount of ROS in the epididymis could not only protect sperm from oxidative damage, but also promote sperm motility [64,65]. Hence, the inhibition of prolactin may affect the function of the epididymis by affecting cAMP concentrations and ROS levels. These changes in the microenvironment of the influencing factors will be further explored in cell experiments.

In recent years, studies have investigated the subtle mechanisms by which the epididymis is involved in not only sperm maturation but also in early embryo development and in the epigenetic transfer of paternal traits to the offspring [66,67,68]. Epigenetic processes, including DNA methylation, chromatin remodeling, histone modifications, and RNA-based mechanisms, can influence gene expression [69]. Fortunately, we found *STPG4* and *POMC* genes, which are involved in paternal epigenetics. The description of the *STPG4* gene at the National Library of Medicine suggests that it is involved in the epigenetic programming of the male pronucleus, and it is predicted to be associated with the positive regulation of DNA demethylation and DNA demethylation in male pronucleus. Epigenetic marks, such as DNA methylation, might modulate *POMC* expression [70]. *POMC* is involved in multiple physiological processes and its epigenetic regulation has been linked to nutritional programming, obesity, energy balance, and metabolic outcomes [70,71]. The above results suggest that PRL might be involved in epididymal sperm maturation through ESR, cAMP, GnRH, ROS, steroid binding, and calcium ion binding pathways, and also associated with epigenetic processes of paternal traits. However, our study was mainly conducted in vivo, and the mechanism of PRL regulation in the epididymis needs to be further explored at the cell level. Based on our RNA-seq results, the regulatory effects of PRL on epididymal epithelial cells through non-coding RNA and small RNA will be investigated, which will provide new insights into the molecular mechanisms of epididymal regulation in bucks.

## 5. Conclusions

Our study confirms that PRL inhibition decreased the expression of PRLR protein and serum E2 concentration and altered the epididymal duct diameter and epididymal epithelium height in Yanshan Cashmere bucks. *ESR2*, *GPER1*, *MAPK10*, *JUN*, *ACTL7A*, and *CALML4* were significantly enriched in multiple pathways, such as ESR, cAMP, GnRH, ROS, and calcium ion binding, and they might be the key genes of PRL in regulating epididymal function.

## Figures and Tables

**Figure 1 animals-14-01778-f001:**
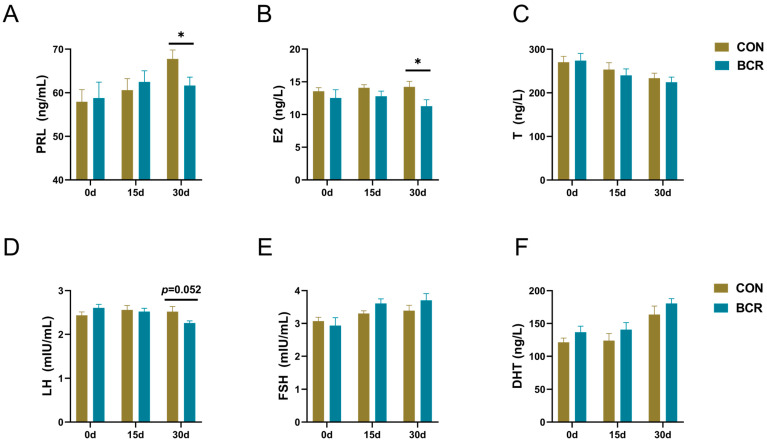
Changes in serum hormone levels in Cashmere bucks. (**A**) PRL, prolactin; (**B**) E2, estradiol; (**C**) T, testosterone; (**D**) LH, luteinizing hormone; (**E**) FSH, follicle-stimulating hormone; and (**F**) DHT, dihydrotestosterone. Values are the mean ± standard error of the mean. BCR, bromocriptine treatment group; CON, control group; * *p* < 0.05.

**Figure 2 animals-14-01778-f002:**
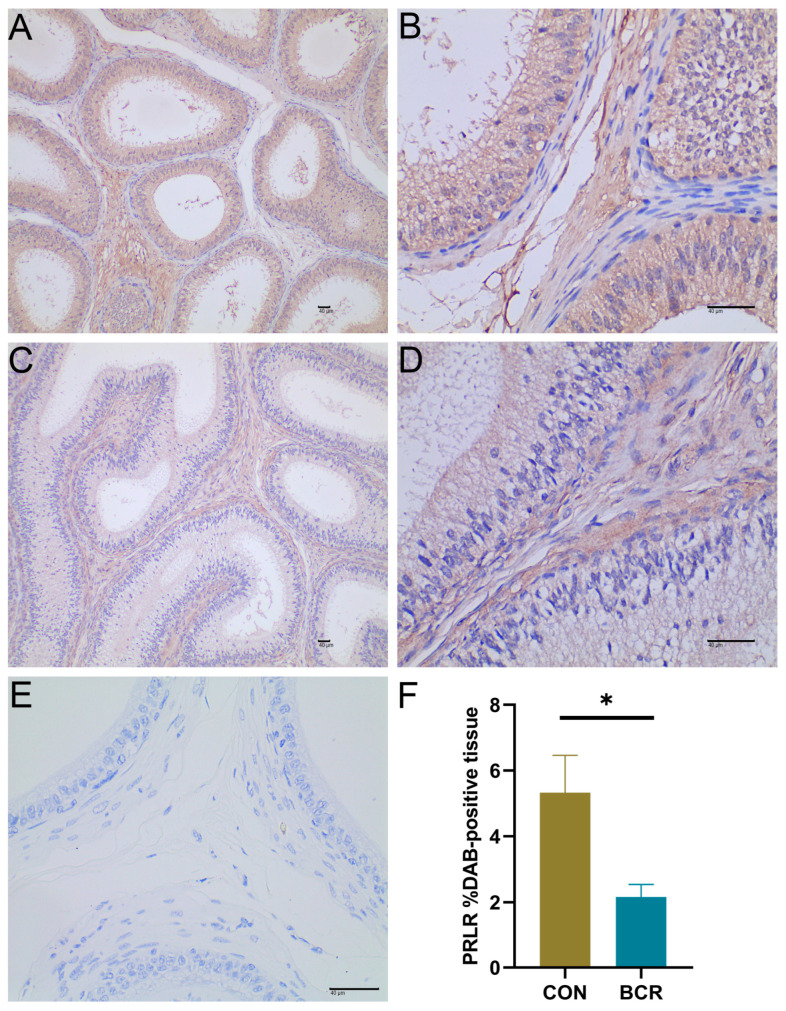
Localization and changes in PRLR proteins in the epididymis of Cashmere bucks. Immunohistochemical staining of epididymal tissues in the (**A**,**B**) CON and (**C**,**D**) BCR groups. (**E**) Negative control section. (**F**) %DAB-positive tissue of the PRLR. (**A**,**C**) Magnification ×100, scale bar = 40 μm. (**B**,**D**,**E**) Magnification ×400, scale bar = 40 μm. Values are the mean ± standard error of the mean (**F**). BCR, bromocriptine treatment group; CON, control group; * *p* < 0.05.

**Figure 3 animals-14-01778-f003:**
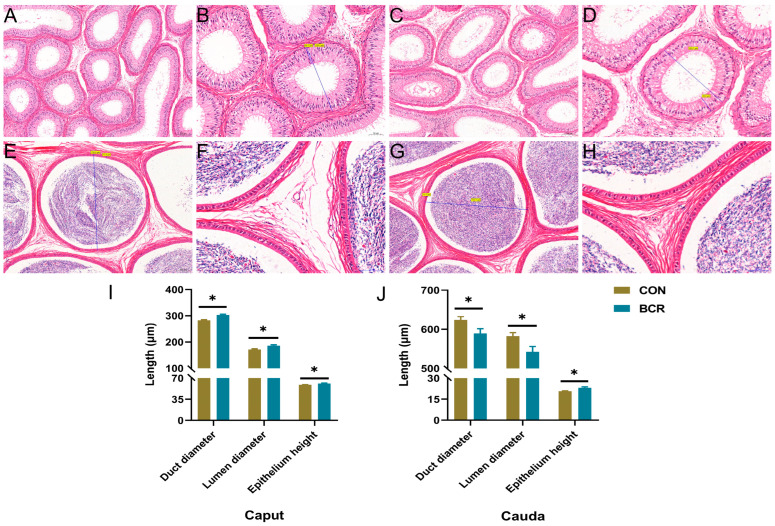
Histologic examination and measurement of the caput and cauda of the epididymis. The CON group (**A**,**B**) and the BCR group (**C**,**D**) in the epididymal caput. The CON group (**E**,**F**) and the BCR group (**G**,**H**) in the epididymal caudal. Epididymal duct diameter, epididymal lumen diameter, and epididymal epithelium height in the caput (**I**) and cauda (**J**) of the epididymis in bucks. Values are the mean ± standard error of the mean. (**A**,**C**,**E**,**G**) Magnification ×100; scale bar = 100 μm. (**B**,**D**) Magnification ×200; scale bar = 50 μm. (**F**,**H**) Magnification ×400; scale bar = 50 μm. Values are the mean ± standard error of the mean. BCR, bromocriptine treatment group; CON, control group; * *p* < 0.05.

**Figure 4 animals-14-01778-f004:**
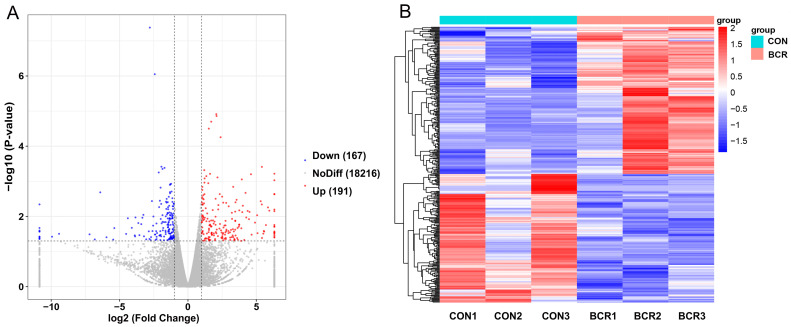
Analyses of differentially expressed genes (DEGs) between the CON and BCR groups. (**A**) Volcano plot of DEGs. Red represents upregulation and blue represents downregulation. (**B**) Hierarchical clustering heatmap of DEGs. BCR, bromocriptine treatment group; CON, control group.

**Figure 5 animals-14-01778-f005:**
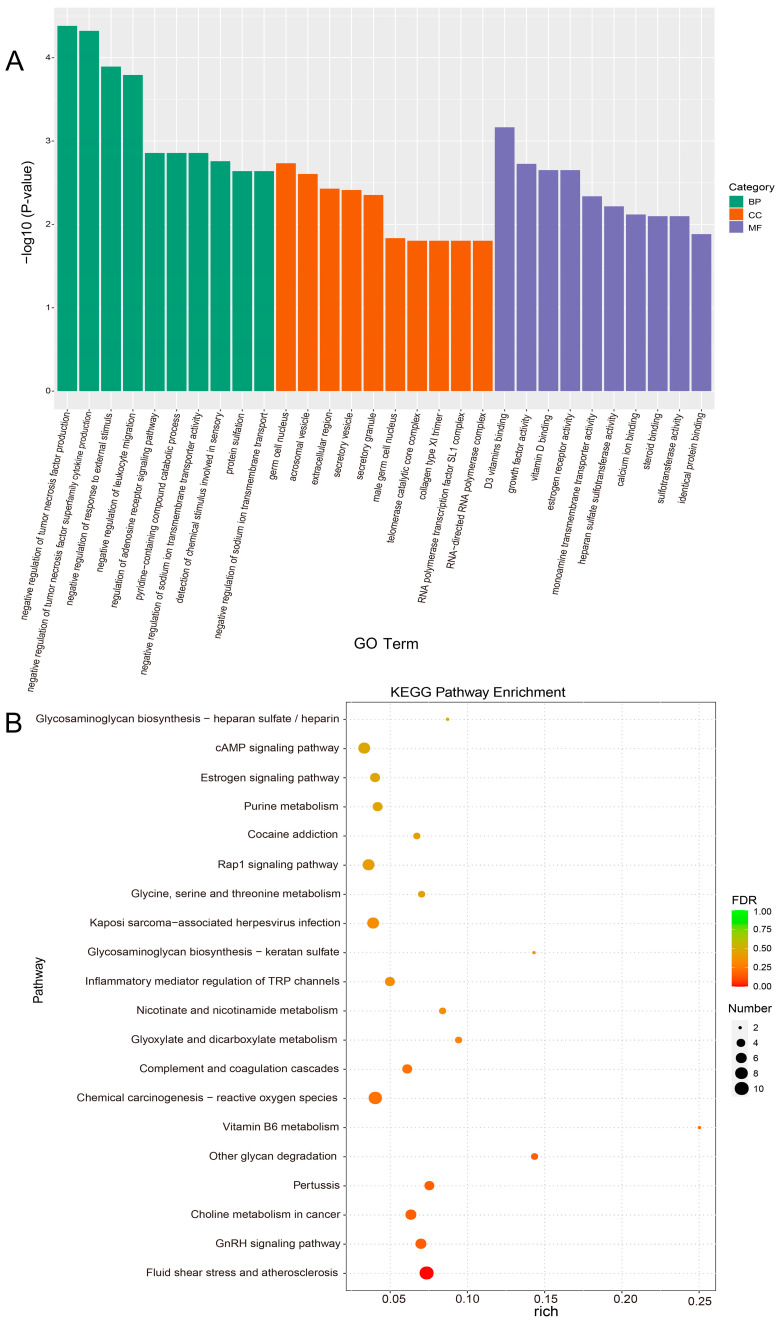
GO and KEGG pathway analyses of differentially expressed genes (DEGs). (**A**) Top 10 enriched GO terms for each of the three categories. BP, biological process; CC, cellular component; MF, molecular function; GO, Gene Ontology. (**B**) Top 20 enriched KEGG pathways. KEGG, Kyoto Encyclopedia of Genes and Genomes.

**Figure 6 animals-14-01778-f006:**
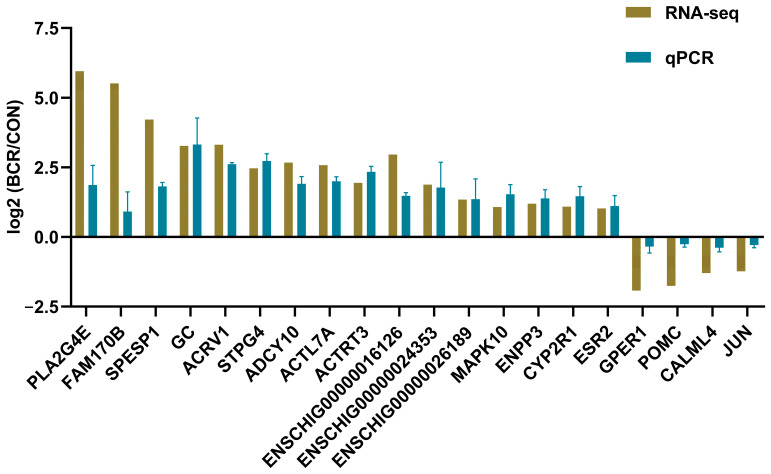
qPCR validation results of the selected 20 differentially expressed mRNAs. Values are the mean ± standard error of the mean. BCR, bromocriptine treatment group; CON, control group.

**Table 1 animals-14-01778-t001:** Primer sequence.

Gene Name	Primer 5′ to 3′		Gene ID
*β-actin*	F	GCGGCATTCACGAAACTACC	102179831
	R	GCCAGGGCAGTGATCTCTTT	
*ACRV1*	F	CAGTTGAACACGCAAGTGGG	102185536
	R	GTTGTTCACCGGAAGTGTGC	
*ACTL7A*	F	GTGTCCTGCCTCAACAAGTG	102184342
	R	GACACATGCTGCTCAGTTCC	
*ACTRT3*	F	TGTGATGGCCCAGTCTTGAT	102181538
	R	AAAGCCAGTTGTGAAGCCAG	
*ADCY10*	F	CTTCAACCTGCCCCTGAAAC	102173054
	R	GATTCCGGAGCAATGACCAC	
*CALML4*	F	TCCCTGTATGACAAGCAGCA	102171428
	R	GTCTATCCTGTGAGTCCGCA	
*CYP2R1*	F	TAATGTGCTACGGTGGGCAAT	102190621
	R	CCTCATGTAAAACTGCTTCGG	
*ENPP3*	F	TGCTCCATTGCCCACAGATA	102169872
	R	GAGACAGGGCTCCTTGTTCT	
*ESR2*	F	AGTGCAAAAATGTGGTGCCC	100860823
	R	GCCCTCTTTGCTCTCACTGT	
*FAM170B*	F	AGAACACCTACTTCTCGGGC	102190381
	R	GCAAGACTGGTACTGGGAGT	
*GC*	F	GCCCAAGGAGCTTCCTGAAT	102176214
	R	CTTTGTTCGTGGGCAACTGG	
*GPER1*	F	CTTCTCCAACAGCTGCCTCA	106503626
	R	CCGGTTTTCTGCTCCAGGTA	
*JUN*	F	CGTCCACTGCCAATATGCTC	102185423
	R	GTTAGCATGAGTTGGCACCC	
*MAPK10*	F	ATGTGGAGAATCGGCCCAAG	102182774
	R	TCGCTGGGTCAATCACTAGC	
*PLA2G4E*	F	AACCTATCCCACACCTCGGA	102187595
	R	ATTTCAGGCAGGAGTGGCTC	
*POMC*	F	GCTGAGCTGGAGTATGGTCT	102182514
	R	GCTCTTCTCCGAGGTCATGA	
*SPESP1*	F	CCAGAGCCAATTGAACCTCG	102171881
	R	GGAACATCTTCTTCCGTGGC	
*STPG4*	F	TTTCCAACAGGCTGCTTCAC	102180769
	R	GCAGGCAAGAATCGAGGTAC	
*ENSCHIG00000016126*	F	CAGGTTTGACCAGGTGACGA	102185549
	R	GAACTGCCATCCCGGAAAGA	
*ENSCHIG00000024353*	F	CCAGGTTGTGTGGAACCAGT	106503207
	R	GTCCGAGTCCTGGAAGTTGG	
*ENSCHIG00000026189*	F	TACGCCTTCTCCCAGTTTCG	102178625
	R	AGAAGGCATACCAGACGTG	

F, forward primers; R, reverse primers.

**Table 2 animals-14-01778-t002:** Detailed information of RNA sequencing.

Sample Name	Raw Reads	Clean Reads	Clean Reads (%)	Q20 (%)	Q30 (%)
CON1	162,455,762	139,289,932	85.74	97.98	94.46
CON2	147,909,398	134,685,882	91.05	97.74	93.77
CON3	146,488,370	133,759,152	91.30	97.67	93.6
BCR1	156,925,208	142,022,078	90.5	97.99	94.17
BCR2	159,208,604	144,256,712	90.6	97.89	93.96
BCR3	154,521,362	141,450,084	91.54	97.76	93.79

BCR, bromocriptine treatment group; CON, control group.

## Data Availability

All datasets generated and analyzed during the current study are available from the corresponding author (zhangyingjie66@126.com) upon reasonable request.

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
