# Peer review of "Inhibition of Prolactin Affects Epididymal Morphology by Decreasing the Secretion of Estradiol in Cashmere Bucks"

_animals, 2024, doi:10.3390/ani14121778_

Round 1
Reviewer 1 Report
Comments and Suggestions for Authors
Referee’s Evaluation Report
MANUSCRIPT IDENTIFICATION: animals-3010010
Prolactin affect epididymal morphology by decreasing the secretion of estradiol
in Cashmere goats
(ORIGINAL ARTICLE)
Comments to Authors/Editor
The paper of Liu & colleagues aimed to investigate the role of prolactin (PRL) in epididymal function in male Cashmere goats, by using bromocriptine, a prolactin inhibitor molecule. Besides, a transcriptomic analysis was performed to unveil differentially expressed genes in treated and control groups. This manuscript falls within the scope of ANIMALS, while it is sufficiently informative for the replication of the study. In general, the organization of the experiment seems to be well designed, yet the sentence structure must be improved. In the title, use Cashmere instead cashmere; correct accordingly the whole manuscript. Simple Summary was written in a careless fashion: L13; … of male cashmere goats or, … of Yanshan-Cashmere bucks??? Also, L15; … as well as significantly changes (what kind of changes?????) the epididymal duct diameter and epididymal epithelium height in goats; please try, … as well as increased -or decreased- (p<0.05 or p<0.01) the epididymal duct diameter and the epithelium height (p<0.01 or p<0.05). Besides, in L16, eliminate “in goats”; the authors already stated in L13, that the study was performed in males Cashmere goats. Moreover, instead the word significantly, use the p-value itself ! ! !; correct accordingly along with the whole manuscript. In the Abstract section, the authors must clarify where and when the samples were collected as well as why they were collected in Yanshan-Cashmere bucks, all for the benefit of the readers of ANIMALS. L31; “change from: … but there were no significant effects, to “no differences (p<0.05) occurred for these variables on the 0th and 15th d”; always avoid the use of “there were no significant differences…”, use “no differences (p>0.05) occurred…”, instead (v.g. L32). L38-39; avoid the use of …”We then identified a total…”; use “Thereafter, a total of 358 differentially expressed genes (DEGs) in epididymal tissues, 191 up-regulated and 167 down-regulated genes were identified”. Correct accordingly along with the whole manuscript. The authors MUST clarify some numbers, especially to include the number of goats in Asia, China, and the region where the samples were collected. Why the authors decided to perform such study in the evaluated breed???, is the Yanshan-Cashmere breed important from a social, productive, or numeric standpoint???; include the body weight of the bucks; please contextualize. How bucks were sampled and what tissues were collected. These issues must be carefully addressed-clarified not only in the Abstract section, but also in the Introduction section. The Introduction section is generally well presented; the authors must contextualize not only the importance of Asia regarding the world goat population, but also another key issues from a social, economic, and productive point of view, especially with respect to fiber-producing breeds. L87; please change from …”We speculate, to “Based on the previous findings, we hypothesize that…”; the objectives of the study were included in the Introduction section. In the Material & Methods section, I do strongly recommend starting with the institutional approval of the study; the authors must state if the study followed any international guide for the use of animals in research; this is a must. The Section 2.1, must include 2.1. Locatio, animals and their management. Please include in this part the location where the study was carried out, including information regarding the latitude, longitude, altitude, prevalent photoperiod and other environmental conditions. Noteworthy, the authors must state if the used bred depicts a seasonal reproduction; this is central to better understand the main reproductive and endocrinological outcomes of the study. L95; “All goats were …, or “All the experimental animals had free access….”???? L99; the authors must include the energy and protein value of the offered diets; this is a must. The authors MUST avoid the use of goats; the study was performed in males, therefore use the term “Bucks, along with the whole manuscript, from the title up to the conclusions. L112-113; do not use r/min, use the g-value. This section must be rewritten; it was written in a carless fashion. 2.3. Hormone analysis or Hormone analyses??? What the authors mean with respect to “The fit indicators????? (L126). In general, reagents, standards, and methods used are relevant and in accordance with the objectives of the study. Nevertheless, I strongly recommend including a figure with the actual experimental protocol across time (i.e., a timeline of actions); this is a must. Section 2.6, Bioinformatics Analysis or Bioinformatic Analyses??? 2.8. Statistical analysis or Statistical analyses???? The experimental design was not clearly explained, while the statistical models were not described for the reader to understand how the experiment was carried out. In the Results Section, as mentioned in the Abstract section, avoid the use of …”There were no differences…”; are the authors comfortable with this grammar structure??? In general, the novelty value of the results is reasonable. In this section the authors presented their main research outcomes in diverse Figures and Tables. Nonetheless, the titles must be rewritten; certainly, the titles of tables must stand by themselves. Moreover, in the subheadings of the graphs A to F, please change from days to d (i.e. 0 days to 0d; 30 days to 30d; the use of “days” again, again, and again in these figures is only “visual contamination”. As mentioned, the authors must avoid the use of both the word “significant” and the “probability value itself”; it is a pleonasm. (i.e., L250-252). Regarding to the Discussion section, I think the way the authors used to initiate this section, was not the best strategy to do. At the beginning of the Discussion, I do strongly suggest initiating this section including the working hypothesis of the study. Authors must define if, with the obtained results, such hypothesis is rejected or non-rejected. For this reason, the authors must include the working hypothesis prior to the objectives in the Introduction section. After this opening paragraph, the authors must follow the same order in this section according to that proposed in the Results section. Moreover, the second paragraph, again, is not the best strategy to continue the Discussion section. The authors must reorganize the structure of the Discussion Section; they must follow the same arrangement and subheading as presented in the Results section; are the authors comfortable with the used approach???? The authors must link, in a logical fashion, their main findings along with the discussion section, to compare & to discuss and, thereafter, be able to propose some possible explanations for such specific outcomes, considering to previous similar studies from the scientific literature. In general, the authors made an accurate interpretation of the main findings. The authors must focus their main findings and confront them with respect to the scientific literature in a logical and focused fashion. In general, the main outcomes of the study were not soundly presented. The list of references cited in the manuscript is proper. This is a very interesting study. Yet, the authors must improve the clarity and logical arrangement of the observed results, especially in the Results and the Discussion sections. The authors must align the conclusions regarding the working hypothesis as well as the scientific question they try to solve; nothing else, just that. All the commented issues and requests should be clearly addressed by the authors. At this point, and based on the above comments, my pronouncement is that this manuscript cannot be accepted in its actual format; it requires major adjustments.
Comments on the Quality of English Language
Already mentioned.
Author Response
Dear Reviewer,
On behalf of my co-authors, we thank you very much for giving us an opportunity to revise our manuscript, and we also appreciate reviewers very much for their positive and constructive comments and suggestions on our manuscript entitled “Inhibition of prolactin affects epididymal morphology by decreasing the secretion of estradiol in Cashmere bucks” (Manuscript Number:3010010).
We revised the manuscript according to these comments and suggestions. In general, we have tried our best to revise our manuscript and provide the point-by-point responses. All changes have been highlighted.
Please see the attachment.
Thank you and best regards.
Sincerely yours,
Yingjie Zhang

Reviewer 2 Report
Comments and Suggestions for Authors
This article tried to describe the inportance of prolactin on epididymal function in goats.
Authors should provide information of ELISAs of PRL, E2, LH, FSH, T and DHT. For PRL, LH and FSH, there are species difference in amino acid sequences. Authors should provide whether these kits are suitable to measure these protein hormones. For steroids, especialy estradiol and testosterone, values seems to be too high. As shown in the reference #26, estrdiol levels are in pg/ml not ng/ml. Testosterone levels are also too high.
Authors should discuss why the PRL levels did not change after 15 days of BCR treatment.
Authors should discuss why the E2 levles decreased on 30 days of BCR treatment.
Author Response

(The authors gave the same response as above.)

Reviewer 3 Report
Comments and Suggestions for Authors
The MS entitled “Prolactin affect epididymal morphology …” written by Liu et al applied techniques including ELISA, HE, IHC, qRT-PCR and RNA-seq to investigate the involvement of prolactin (PRL), by its inhibitor bromocriptine, in the epididymal function of male cashmere goats, providing new insights into the molecular mechanisms for PRL regulating the epididymis function in goats.
General comment
The authors found that inhibition of prolactin decreased serum estradiol concentrations and expression of prolactin receptor protein in the epididymis, as well as alterations the epididymal duct diameter and epithelium height in goats, which might be through the differentially expressed genes including ESR2, MAPK10, JUN, ACTL7A, and CALML4. This research contributes to a better understanding of the mechanisms governing physiological changes in the epididymis. The area of investigation is interesting, but the manuscript contained few flaws which might affect the quality of the publication in this journal.
Specific comment
Title needs to change due to that the inhibitor of PRL, not PRL, decreased the secretion of E2.
P95: Regarding to the feeding, how about the roughage? Such as the composition of the roughage. How to feed goats? Such as the time, the order of feeding
P112: 3000 r/ use g for the unit
P116: A portion of the right epididymis tissue
P167: Total RNA was isolated from epididymal tissues
It is not clear, such as tissues from which part of the epididymis? Noted that the epididymis includes Caput, Corpora, Capda, and the effect of PRL for the Caput and Capda is a little different as shown in the HE by authors, which is also needed to be explained in the discussion part.
P222:There is lack of writing the method for Real Time PCR as per the MIQE guidelines. It is unclear for the Amount of RNA and reaction volume, Reverse transcriptase and concentration, PCR efficiency calculated from slope, pls prepare based on the MIQE
P240: Fig. 1 the unit ml should be mL; use Day in the title of the X axis
P325: best to provide data as the mean ±SE
Comments on the Quality of English LanguageMinor editing of English language required.
Author Response

(The authors gave the same response as above.)

Round 2
Reviewer 1 Report
Comments and Suggestions for Authors
As I mentioned in the first evaluation of the original version and regarding to the Discussion section, I think the way the authors used to initiate this section, was not the best strategy to do. At the beginning of the Discussion, I do strongly suggest initiating this section including the working hypothesis of the study. Authors must define if, with the obtained results, such hypothesis is rejected or non-rejected. For this reason, the authors must include the working hypothesis prior to the objectives in the Introduction section. After this opening paragraph, the authors must follow the same order in this section according to that proposed in the Results section. Please follow the advise and include the suggested opening paragraph, based on the working hypothesis.
Author Response
Dear Editor and Reviewers,
On behalf of my co-authors, we thank you very much for giving us an opportunity to revise our manuscript, and we also appreciate reviewers very much for their positive and constructive comments and suggestions on our manuscript entitled “Inhibition of prolactin affects epididymal morphology by decreasing the secretion of estradiol in Cashmere bucks” (Manuscript Number:3010010).
We revised the manuscript according to these comments and suggestions. In general, we have tried our best to revise our manuscript and provide the point-by-point responses. All changes have been highlighted.
Once again, thank you very much for your comments and suggestions. And we hope that the revised manuscript can be accepted by Animals. If further revision is necessary, please contact me at: zhangyingjie66@126.com and liuxiaonahau@163.com.
Please see the attachment.
Thank you and best regards.
Sincerely yours,
Yingjie Zhang

Reviewer 2 Report
Comments and Suggestions for Authors
Authors should provide enough information to repeat and confirm the research in the manuscript. For the hormone analyses, no information was provided for each ELISA, such as product numbers. What are batch numbers of goat ELISA kits, for PRL and LH? Are there any web site for these hormone kits? If not, authors should provide supplementary files of detailed protocol for each hormone.
Author Response

(The authors gave the same response as above.)
